# Ultra-Wide-Field Fluorescein Angiography Assessment of Non-Perfusion in Patients with Diabetic Retinopathy Treated with Anti-Vascular Endothelial Growth Factor Therapy

**DOI:** 10.3390/jcm12041365

**Published:** 2023-02-08

**Authors:** Jean-Baptiste Morel, Franck Fajnkuchen, Fatima Amari, Nanthara Sritharan, Coralie Bloch-Queyrat, Audrey Giocanti-Aurégan

**Affiliations:** 1Ophthalmology Department, Paris Seine Saint Denis Hospital, Sorbonne Paris Nord University, 125 Rue de Stalingrad, 93000 Bobigny, France; 2Department of Clinical Research, Paris Seine Saint Denis Hospital, Assistance Publique-Hôpitaux de Paris (AP-HP), 93000 Bobigny, France

**Keywords:** diabetic retinopathy, ischemia, fluorescein angiography, diabetic macular edema, anti-VEGF therapy

## Abstract

**Purpose**: To follow the evolution of peripheral ischemia by fluorescein angiography (FA) on ultra-wide-field (UWF) images in diabetic patients treated with anti-vascular endothelial growth factor (anti-VEGF) for macular edema. **Methods**: Prospective, non-interventional cohort study analyzing UWF-FA images of 48 patients with diabetic retinopathy (48 eyes) treated for diabetic macular edema. UWF-FA was performed at baseline and after one year of anti-VEGF therapy (M12). The primary endpoint was the change in the non-perfusion index. **Results**: Of the 48 patients included in this study, 25 completed the one-year follow-up, and 20 had FA images of sufficient quality to be interpreted. The non-perfusion index did not significantly change from baseline after one year of anti-VEGF treatment (0.7% of the non-perfused area at baseline versus 0.5% at M12; *p* = 0.29). In contrast, the diabetic retinopathy severity score improved significantly between baseline and M12. **Conclusions**: Anti-VEGF treatment with aflibercept for diabetic macular edema had no impact on the retinal perfusion assessed by FA, but it allowed for artificially improving diabetic retinopathy severity scores.

## 1. Introduction

Diabetic macular edema (DME) is a common microvascular complication of type 1 and 2 diabetes and a leading cause of visual impairment [1,2,3]. Anti-vascular endothelial growth factor (VEGF) injections are generally used as a first-line therapy for DME to improve visual acuity [4,5].

Anti-VEGF injections used for DME treatment lead to a reduction in diabetic retinopathy (DR) stages assessed by Early Treatment of Diabetic Retinopathy Study (ETDRS) seven-standard-field color fundus photography [6]. This improvement in the diabetic retinopathy severity scale (DRSS) score is based on indirect signs of retinal ischemia (hemorrhages, micro-aneurysms, and retinal microvascular abnormalities). Fluorescein angiography (FA) allows for directly assessing the retinal non-perfusion.

Optos California v2.14 imaging system (Optos, Scotland) allows obtaining with only one picture an ultra-wide-field (UWF) acquisition of 200° of the retina, which is clearly larger than ETDRS seven-standard fields and includes a FA module [7,8]. A previous study using this system has suggested that predominant peripheral lesions that cannot be seen on the ETDRS seven-standard fields could be important predictors of DR progression [9]. Thus, this system could improve DR severity assessment.

The aim of this study was to follow the change in retinal non-perfusion by UWF-FA in diabetic patients treated with anti-VEGF (aflibercept) injections for DME.

## 2. Materials and Methods

### 2.1. Study Population

In this prospective cohort study, consecutive diabetic patients with vision loss due to treatment-naïve DME were included from April 2017 to June 2019, with a follow-up of 12 months. Inclusion criteria were patients with diabetes (Type 1 or 2) aged 18 years or older, with at least one eye with a best visual acuity ranging between 5 and 78 ETDRS letters due to DME (with a central macular thickness [CMT] > 310 µm), requiring aflibercept treatment administered according to the official recommendations, registered with the French social security system. Only patients with UWF-FA images of sufficient quality and who consented to participate in the study were included. Exclusion criteria were patients with an allergy to aflibercept, having received previous intravitreal anti-VEGF therapy in the last 12 months, with proliferative DR, or with a history of pan-retinal photocoagulation.

This study was approved by the Institutional Ethics Committee of Avicenne Hospital and adhered to the tenets of the Declaration of Helsinki. Informed consent was obtained from all patients.

### 2.2. Outcomes

The main outcome of this study was the change in the non-perfusion index between baseline and M12. The secondary outcomes were the ETDRS score assessed on the ETDRS seven-standard-field color fundus images and on UWF images, the change in predominant peripheral lesion number, the change in best-corrected visual acuity (BCVA) on the ETDRS scale between baseline and M12, the change in CMT on Spectral Domain-Optical Coherence Tomography (SD-OCT) between baseline and M12 and the number of aflibercept intravitreal injections received during the first year of DME treatment.

### 2.3. Retinal Image Acquisition

UWF color photography was performed using the Optos California v2.14 imaging system (Optos, Scotland) at baseline, M3, M6, M9, and M12. UWF-FA images were obtained using the same system after intravenous administration of fluorescein. Images were captured in the early (45 s), middle (2 min), and late (5 min) phases of FA at baseline and M12.

The CMT was measured on the SD-OCT B-scan using the Triton system (Topcon, Japan).

All acquisitions were performed in the ophthalmology department of Avicenne Hospital. Images were anonymized and identified by a code.

Images were exported in JPEG format and then automatically aligned for each patient using i2kRetina software (DualAlign, Clifton Park, NY, USA) and cropped to keep only the part of both images common between baseline and M12. After alignment, UWF-FA images were split into 16 identical boxes to facilitate the search for areas of ischemia.

### 2.4. Quantitative and Qualitative Analyzes

UWF-FA and color images obtained at baseline and M12 were presented in a random order without providing any indication of the time the examination was performed to two retina specialists experienced in DR grading. For each image, the DRSS score was assessed based on the ETDRS score and on the simplified American Academy of Ophthalmology (AAO) DR grading scale using a five-stage disease severity grading, first on the ETDRS seven-standard fields, and then on UWF-FA. The predominant peripheral lesions (punctiform hemorrhages, microaneurysms, intraretinal microvascular abnormalities (IRMAs), and spot or flare hemorrhages) were counted (Figure 1).

Only the blocks analyzable at baseline and M12 were analyzed. In this figure, five blocks out of 16 were excluded, i.e., 31%. The average analyzable block for the 20 patients in the study is 49%. The number of predominantly peripheral lesions is noted in each block.

Regarding UWF-FA images, non-perfusion was defined as a fundus area devoid of retinal arterioles, venules, and/or capillaries, with a “pruned” appearance of adjacent vessels. The boxes with non-perfusion were counted on each image, and then the non-perfusion area was measured in each box using ImageJ software (National Institutes of Health, Bethesda, MD, USA) and related to the whole retinal area analyzed to obtain the non-perfusion index. Each image was independently analyzed side by side by two readers. Discrepancies between the two readers (Intergrader agreement for detection of the ischemic area was low, with a kappa coefficient = 0.4) were resolved by common agreement (Figure 2).

### 2.5. Statistical Analyses

The baseline and follow-up data collected are described as numbers of patients and percentages for categorical variables and as medians and interquartile ranges [IQR] or mean and standard deviation (SD) for quantitative variables.

The analysis population was patients who had angiographic images of sufficient quality to be interpreted.

The primary endpoint (i.e., the non-perfusion index) was analyzed using a Student’s t-test for paired data. The secondary endpoints were analyzed using the same method as the primary endpoint and described according to their distribution and nature. Data transformations have been performed for all endpoints except for the central macular thickness in order to respect the conditions of application of the test (ordered quantile normalization transformation). ETDRS score was assessed with UWF, and ETDRS seven-standard field data were analyzed using a mixed-effect model for ordinal data. The odds ratio (OR) and their 95% CIs were reported. No imputation method was used for missing data, and each endpoint was analyzed in patients with baseline and M12 available data. All statistical analyzes were performed using R software (version 3.5.2). A *p*-value < 0.05 was considered significant. Fleiss’ Kappa statistic was calculated to assess inter-grader agreement.

## 3. Results

Forty-eight patients were included in the study between April 2017 and June 2019. Among them, 25 completed the one-year follow-up, and 20 had FA images of sufficient quality to be interpreted (Figure 3).

Patients’ mean age (±SD) was 64.7 ± 10.0 years, and 65.6% were men. Also, 95.8% of patients had type 2 diabetes, and 39.6% and 60.4% had, respectively, moderate and severe non-proliferative DR.

Table 1 summarizes the baseline characteristics of the patients. Appendix A summarizes Patients’ baseline demographics of analysis population and population not included in the analysis.

For the primary endpoint, there was no significant change in the non-perfusion index between baseline and M12, with a median [IQR] percentage of non-perfused area ranging between 0.7% [0.2; 2.5] and 0.5% [0.0; 1.3] (*p* = 0.29) (Figure 4).

There was no significant change in BCVA that remained stable, with a median BCVA of 70 ETDRS letters and a median difference [IQR] of −5.0 [−15.0; 7.5] ETDRS letters (*p* = 0.4). The median CMT [IQR] significantly improved from 405.0 µm [327.8; 476.5] at baseline to 293 µm [263.2; 365.5] at M12 (*p* = 0.005).

Regarding the ETDRS score for DR severity, there was a significant decrease in this score both on the ETDRS seven-standard fields (OR [IC95%]: 0.2 [0.1; 0.9], *p* = 0.04) and on UWF images (OR [IC95%]: 0.01 [0.0002–0.4]; *p* = 0.02). There was also a significant decrease in the number of predominantly peripheral lesions between baseline and M12, with a median decrease [IQR] from 5.0 [2.0; 10.0] to 3.0 [1.0; 5.0] (*p* = 0.002).

Patients received a mean number of 6.5 ± 2.0 intravitreal injections during the one-year follow-up.

Table 2 and Table 3 summarize the main results of the study.

## 4. Discussion

In our study, using the non-perfusion index, we showed the absence of improvement in retinal non-perfusion assessed by UWF-FA between the initial examination and M12 in patients with non-proliferative DR and DME, treated with anti-VEGF for one year. We chose to quantitatively analyze retinal non-perfusion using the non-perfusion index corresponding to the non-perfused area/total retinal area assessed ratio. The use of this index seemed to be the most adapted solution to highlight a change in peripheral ischemia assessed by FA. Other authors, such as Bonnin et al. [10] and Couturier et al. [11], have qualitatively assessed the retinal non-perfusion without quantitative measurements. Others, such as Levin et al. [12], have chosen to convert the ImageJ angiography images into grayscale and then define as ischemic areas all zones with an intensity of less than 35% on the grayscale.

As previously suggested by Bonnin et al. [10] and Couturier et al. [11], peripheral retinal ischemia does not seem to be improved by anti-VEGF treatment, which only induces the disappearance of indirect signs of ischemia such as retinal hemorrhages, venous abnormalities, or IRMAs. However, these results contrast with those by Levin et al. [12], who have described an improvement in ischemic area reperfusion in 12 out of 16 treated eyes (75%), in a retrospective study, in patients treated with anti-VEGF for DME in the context of non-proliferative or proliferative DR. This difference between the studies could be explained by some properties of anti-VEGF agents. Indeed, anti-VEGFs such as aflibercept used in our study, through their anti-angiogenic effect, reduce the vascular parietal diffusion of fluorescein [13,14,15,16,17,18]. This positive effect on permeability could also improve the contrast between the black of the non-perfused areas, the light gray of the perfused retina and the white of the retinal vessels seen with the Optos California device. The method used by Levin et al. [12] to assess ischemia could have been impacted by this property of anti-VEGFs, affecting the analysis of ischemic areas by FA on the gray scales of ImageJ software.

In our study, the median non-perfusion index was very low compared to what was expected in patients with moderate or severe non-proliferative DR. Indeed, Borrelli et al. [19] in their study in six eyes of patients with non-proliferative DR treated with intravitreal dexamethasone implants for DME, have found a mean (±SD) baseline non-perfusion index of 0.27 ± 0.14 while in our study, the median percentage was lower with 0.7% at baseline and 0.5% at M12. This difference could be explained by our method of analysis: very peripheral blocks were often excluded from the analysis, which was not the case in the study by Borrelli et al. [19]. Moreover, ischemia was mainly located in the very peripheral retina in the examples presented in the study by Borrelli et al. [19].

In our study, we found a significant improvement in DR score with an equally significant decrease in predominantly peripheral lesion numbers. This result is in line with previous studies and confirms that despite the small difference in the non-perfusion index in the blocks analyzed, there was a significant effect of anti-VEGF between baseline and M12 in the blocks analyzed for DR severity score.

In the VIVID and VISTA randomized controlled trials [20], the DR severity score improved significantly in the aflibercept groups compared to the laser control group. In the VISTA trial, this improvement was 33% and 29% versus 14% in the control group (*p* < 0.01), while in the VIVID trial, it was 33% and 27% versus 7% in the control group (*p* < 0.001).

Based on previous results on the stability of ischemia under aflibercept treatment, this improvement in DR score and the decrease in predominantly peripheral lesion number seen on color retinography suggest that monitoring DR by color retinography is insufficient in patients treated with anti-VEGF.

Anti-VEGF agents such as aflibercept used in our study, through their anti-angiogenic effect, allow the resolution of indirect signs of retinal ischemia such as hemorrhages and decrease the parietal vascular diffusion of fluorescein but do not allow for reducing retinal non-perfusion areas. The study of ischemic areas by FA could therefore be impaired by the anti-VEGF treatment rather than supporting the use of OCT-Angiography (OCT-A) for the assessment of non-perfusion. Indeed, OCT-A, a recent non-invasive technique for imaging the retinal vasculature, is becoming increasingly important in the routine management of patients at the expense of FA. Cui et al. [21] have proposed the combined use of UWF retinography and wide-field OCT-A as an alternative to UWF-FA for the detection of DR lesions, with similar detection of microaneurysms, IRMAs, non-perfusion areas, pre-retinal and pre-papillary neovessels for less invasive management (*p* > 0.005).

Russel et al. [22] have suggested in their study comparing wide-field OCT-A to UWF-FA that wide-field OCT-A alone could be sufficient for the diagnosis of proliferative DR with the detection of 99% of neovessels seen on UWF-FA. In a second study, they have even proposed a new staging of DR based on the sole use of wide-field OCT-A.

However, although sometimes considered as an outdated and invasive examination with a non-negligible risk [23], FA, especially when combined with a UWF analysis, remains of definite interest for the diagnosis and follow-up of DR. Indeed, with a faster acquisition time than wide-field OCT-A, and a lesser need for patient cooperation, UWF-FA allows not only analyzing the peripheral ischemic retina but also central ischemic areas on a single image, with easy visualization of pre-retinal neovascularization which is a feared complication that could be challenging to detect on simple color retinography.

In our study, despite a significant decrease in CMT, the visual acuity did not significantly improve in contrast with the results of similar studies. The median value remained stable despite a slight improvement in visual acuity. Korobelnik et al. [20] have observed a gain of 12 and 10 ETDRS letters in the VISTA study at 52 weeks versus 0.2 ETDRS letters in the control group [*p* < 0.0001], and a gain of 10 and 10 ETDRS letters versus 1 ETDRS letter in the control group in the VIVID study [*p* < 0.0001]. The lack of significance of this result in our study could be explained by several factors. On the one hand, the number of patients was smaller than in the randomized controlled trials, so a larger difference in absolute value was needed to achieve significance, and on the other hand, the median baseline visual acuity in our study was higher than in the comparative studies, which limited the potential gain. Indeed, at baseline, the median visual acuity was 70 ETDRS letters in our study, compared to 64 ETDRS letters in the study by Couturier et al. [11], 54–57 ETDRS letters in the RISE and RIDE studies [24], and 58–60 ETDRS letters in the VIVID and VISTA studies [20]. Moreover, it is known that in certain situations, a normal CMT may be found without improvement in visual acuity: this is the case, for example, when disorganization of inner retinal layers (DRIL) is present [25,26,27,28].

In addition, the mean number of intravitreal injections in our study was limited to 6.5, which was lower than what was planned in the protocol and could also explain the disappointing visual outcomes. In fact, according to current recommendations, the induction treatment regimen used in the study was five monthly injections, followed by a series of four bimonthly injections: i.e., nine injections over the first year of DME treatment. It could be assumed that the actual number of injections received was sufficient to achieve the anatomical effects on the CMT and the improvement in the DR score without resulting in a functional improvement in visual acuity. In their study of 18 patients treated with anti-VEGF for DME, lost to follow-up for one year and then followed again, Kim et al. [29] have shown that after treatment resumption, a normal CMT could be achieved in the absence of recovery of the initial visual acuity. Thus, the discontinuous treatment received by our patients could also explain the poorer recovery of visual acuity observed compared to other studies.

One of the strengths of our study was its prospective design, with a long follow-up of 12 months, including five visits for data collection. The prospective design of the study reinforces the validity of our results and explains a large number of patients lost to follow-up, contrary to retrospective studies such as those by Bonnin et al., Couturier et al., and Levin et al.

However, our study has several limitations. It is a monocentric study, with a high rate of patients lost to follow-up who did not complete a full year of follow-up. This high rate confirms the difficulty for diabetic patients to adhere to their management plans. These patients with a disabling disease, already at the microangiopathy stage if they have DR, must be followed by multiple specialists for treatment of the complications related to these chronic diseases. Better consideration of these difficulties, with the use of strategies to promote compliance in diabetic patients, could improve the visual gain under anti-VEGF therapy in these patients.

This low compliance has already been noted by Stéphan et al. [30] in their targeted population and reminds us of the importance of adjusting the follow-up by intensifying therapeutic discussion and counseling in patients at risk of poor compliance.

In conclusion, anti-VEGF treatment with aflibercept for DME does not improve retinal non-perfusion, although it allows for artificially improving the DR severity score on color retinography.

Closer monitoring of DR is recommended, particularly if anti-VEGF therapy is discontinued in patients treated for DME, and the use of UWF-FA seems to be a reasonable option.

## Figures and Tables

**Figure 1 jcm-12-01365-f001:**
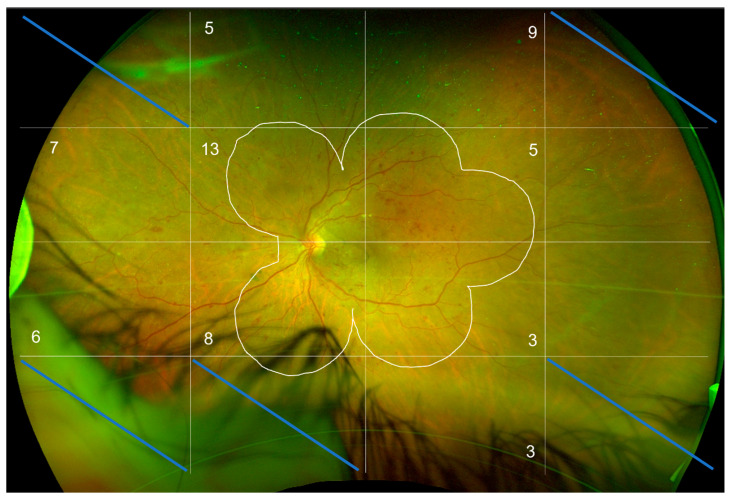
R/G UWF retinography of a left eye at baseline. The image was aligned with the image obtained at M12 and then divided into 16 blocks to facilitate the counting of peripheral lesions located outside the ETDRS 7-standard fields (white line). The numbers in the blocks stand for the number of peripheral lesions in each block outside of the ETDRS7-standard fields.

**Figure 2 jcm-12-01365-f002:**
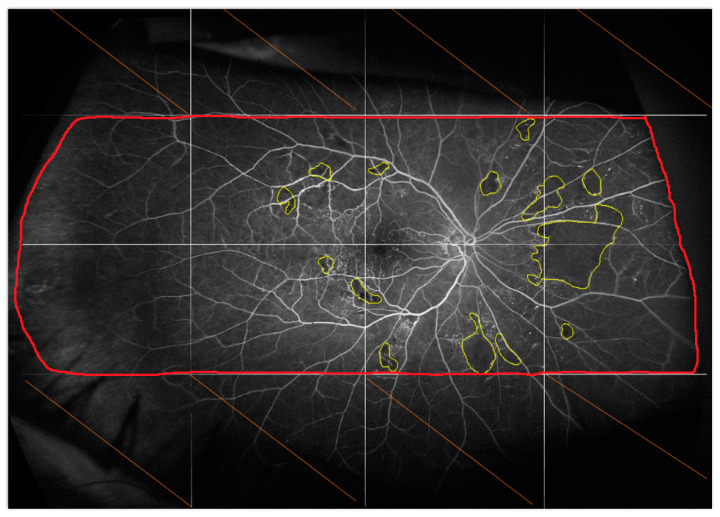
Fluorescein angiogram of the right eye acquired 45 s after dye injection at baseline. The image was aligned with the image obtained at M12 and then divided into 16 blocks to facilitate the identification of ischemia. Only the blocks analyzable at baseline and M12 were analyzed.

**Figure 3 jcm-12-01365-f003:**
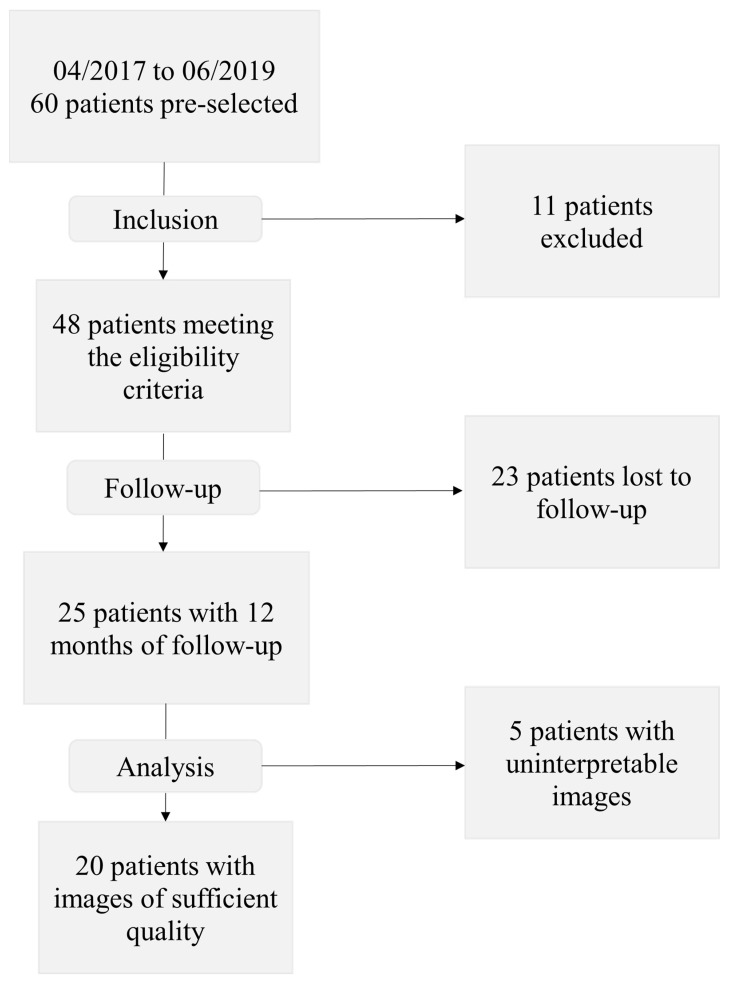
Study flowchart.

**Figure 4 jcm-12-01365-f004:**
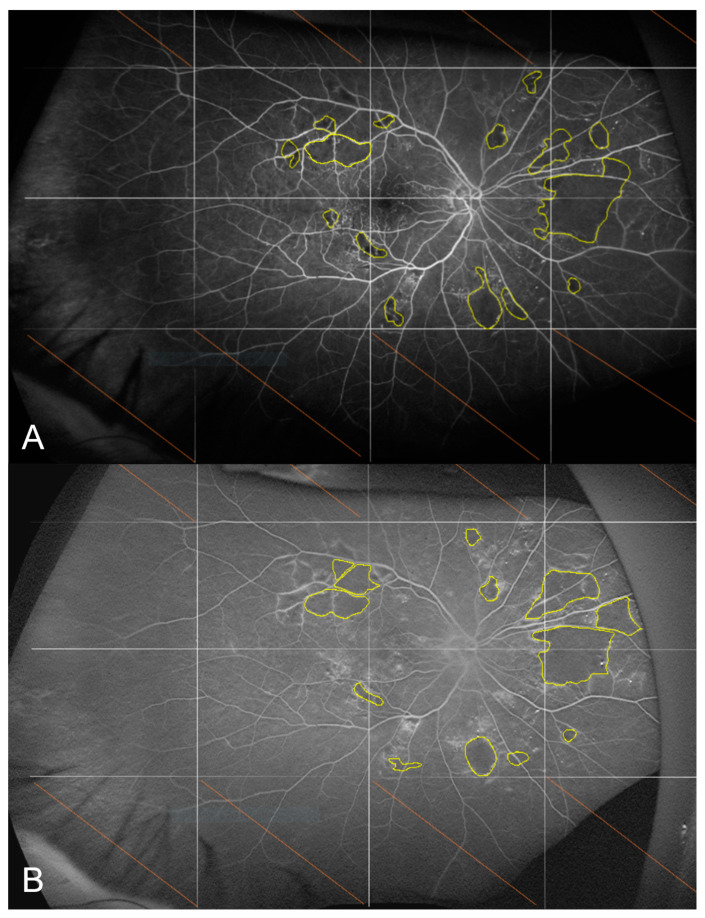
Fluorescein angiograms of the right eye acquired 45 s after dye injection at baseline (**A**) and at M12 (**B**). The non-perfusion index was 0.6% at M0 and 0.7% at M12 for this patient.

**Table 1 jcm-12-01365-t001:** Patients’ baseline demographics.

Variable	Total
	N = 48
Mean age, years, (±SD)	64 ± 10
Male gender, *n* (%)	31 (65%)
Right eye, *n* (%)	27 (56%)
Median BCVA, ETDRS letters [IQR]	68 [50; 75]
Median CMT on OCT, µm, [IQR]	399 [339; 468]
Type 1 diabetes, *n* (%)	2 (4%)
Median Hba1c, % [IQR]	8 [7; 9]
Patients treated with insulin, *n* (%)	29 (60%)
Macroangiopathy, *n* (%)	9 (19%)
DR stage, *n* (%)	
Moderate	19 (40%)
Severe	29 (60%)
Previous treatment for DME, *n* (%)	
none	45 (94%)
anti-VEGF	2 (4%)
corticosteroids	0 (0%)
laser	1 (2%)
Previous PRP, *n* (%)	0 (0%)
Lens status: phakic, *n* (%)	25 (52%)
HBP, *n* (%)	21 (45%)
Renal status, *n* (%)	
No renal failure	38 (81%)
Microalbuminuria	3 (6%)
Renal failure	6 (12%)
Dyslipidemia, *n* (%)	20 (43%)
Sleep apnea, *n* (%)	2 (4%)
OHT or glaucoma, *n* (%)	5 (10%)

SD: standard deviation, IQR: interquartile range, *n*: number, BCVA: best-corrected visual acuity, CMT: central macular thickness, µm: micrometers, HbA1c: glycated hemoglobin, DR: diabetic retinopathy, DME: diabetic, macular edema, VEGF: vascular endothelial growth factor, PRP: pan-retinal photocoagulation, HBP: high blood pressure, OHT: ocular hypertonia.

**Table 2 jcm-12-01365-t002:** Summary of the main results.

Variable		OR [95% CI]	*p-*Value
ETDRS score (ETDRS 7-standard fields), letters N = 17	Baseline	0.23 [0.06; 0.93]	0.04
	M12		
ETDRS score (UWF), letters N = 17	Baseline	0.01 [0.0002; 0.43]	0.02
	M12		

M12: month 12, CMT: central macular thickness, µm: micrometers, PPL: predominantly peripheral lesions.

**Table 3 jcm-12-01365-t003:** Changes in ETDRS score between baseline and month 12.

Variable		Median [IQR]	Median Difference [IQR]	*p-*Value Paired Student’s *t*-Test
Non-perfusion index %, N = 20	Baseline	0.7 [0.2; 2.5]	0 [−0.3; 0.3]	0.29
	M12	0.5 [0; 1.3]		
Visual acuity, ETDRS letters N = 19	Baseline	70.0 [55.0; 75.0]	−5.0 [−15.0; 7.5]	0.41
	M12	70.0 [67.5; 77.5]		
CMT (OCT), µ, N = 14	Baseline	405.0 [327.8; 476.5]	55.0 [27.3; 154.3]	0.005
	M12	292.5 [263.2; 365.5]		
PPL N = 17	Baseline	5.0 [2.0; 10.0]	2.0 [0.0; 8.0]	0.002
	M12	3.0 [1.0; 5.0]		

M12: month 12, ETDRS: early treatment diabetic retinopathy study, UWF: Ultra-Wide Field.

## Data Availability

Data are available upon request by sending an email at audrey.giocanti@aphp.fr.

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
