# Peer review of "Ultra-Wide-Field Fluorescein Angiography Assessment of Non-Perfusion in Patients with Diabetic Retinopathy Treated with Anti-Vascular Endothelial Growth Factor Therapy"

_jcm, 2023, doi:10.3390/jcm12041365_

Round 1
Reviewer 1 Report
1) Line 103-104, only the blocks analyzable at baseline and M12 are analyzed. What percentage of blocks are analyzable? I wonder if this exclusion and inclusion of data were not random and would cause intrinsic bias in the final data included.
2) There is no control group in this study by design. What is the non-perfusing index in a group of patients who have similar characteristics but no diabetic retinopathy? It is assumed in this study that the non-perfusion is due to diabetic retinopathy, but this assumption might be false in patients with other diseases that compromises microvasculature such as thromboembolic events and atherosclerosi
3) Line 185-188: these look like someone’s guidance on how to write a good discussion? Was this part included by mistake?
4) The percentage of the non-perfusion area is very low in the cohort in this study compared to other similar studies. Authors provided explanation that the very peripheral blocks were excluded. What is the rationale for excluding the very peripheral blocks? Is there any standards or criteria guiding such exclusion? As mentioned in line 221, most of the ischemic area are located in the very peripheral retina, then it is reasonable to believe that, if there are going to be any chances in the non-perfusion area, these changes will most likely happen to/be detected at the very peripheral of the retina. Such exclusion might cause bias. Author should be encouraged to provide the specific reason and criteria how the exclusion was made.
5) Out of 48 patients, only 25 completed all follow up, and only 20 had FA images that can be used for this study: this high percentage of loss of follow up can cause selection bias. What are the reasons for lost follow up? It is not safe to assume this loss of follow up is random, thus causing bias in the data included. Specifically, in a scenario where the patients who loss follow up because they became more visually impaired to attend appointments, such loss of follow up will neglect a lot of data on people who have worse clinical outcomes, causing bias that is significant enough to compromise the validity of the study.
Such concern is not ungrounded: the patient cohort included in this study seems to be healthier in terms of the vision compared to other studies, evident by 1) better baseline vision of 70 ETDRS letters (compared to others at 50s) and smaller percentage of non-perfusion areas. This makes me worry about the generalizability of the study results. Also the number of patients is small.
6) Some minor grammar mistakes: eg 253-254 UWF-FA allows analyzing [not only] the peripheral ischemic retina but also central ischemic area on a single image.
7) Could the baseline characteristics of the 20 interpretable patients be added to the study? Is there any similar analysis done for these 20 patients’ data only (baseline vs M12)? How does that result look like? Given that data of more than half of the original 48 patients were not used in analysis, it remains in question how meaningful of an interpretation can be generated based on the baseline characteristic of all 48 patients.
Author Response
1) Line 103-104, only the blocks analyzable at baseline and M12 are analyzed. What percentage of blocks are analyzable? I wonder if this exclusion and inclusion of data were not random and would cause intrinsic bias in the final data included.
The blocks were excluded from the analysis if the visualization of the block does not allow to analyze ischemia due to the presence of eyelashes for example. In figure1, 5 blocks out of 16 were excluded, i.e. 31%. The average analyzable blocks for the 20 patients in the study is 49%, or almost 8 out of 16 blocks for each patient. We have added this information in the method section.
2) There is no control group in this study by design. What is the non-perfusing index in a group of patients who have similar characteristics but no diabetic retinopathy? It is assumed in this study that the non-perfusion is due to diabetic retinopathy, but this assumption might be false in patients with other diseases that compromises microvasculature such as thromboembolic events and atherosclerosis?
Indeed our study does not include a control group in its design as it is also the case in similar studies. It would be interesting to have data on retinal ischemia assessed by fluorescein angiography in only hypertensive patients for example. However, since angiography remains an invasive examination with a significant risk of severe allergic reaction, this study cannot be justified from an ethical point of view in its prospective design. Moreover, our study focused on the evolution of the ischemic index throughout one year of treatment by antiVEGF more than the rate per se, the question was does antiVEGF improve or not the perfusion of the retina ?
3) Line 185-188: these look like someone’s guidance on how to write a good discussion? Was this part included by mistake?
The lines have been deleted.
4) The percentage of the non-perfusion area is very low in the cohort in this study compared to other similar studies. Authors provided explanation that the very peripheral blocks were excluded. What is the rationale for excluding the very peripheral blocks? Is there any standards or criteria guiding such exclusion? As mentioned in line 221, most of the ischemic area are located in the very peripheral retina, then it is reasonable to believe that, if there are going to be any chances in the non-perfusion area, these changes will most likely happen to/be detected at the very peripheral of the retina. Such exclusion might cause bias. Author should be encouraged to provide the specific reason and criteria how the exclusion was made.
Indeed, the blocks excluded were mainly peripheral blocks because these blocks are the first affected by artifacts on UWF retinograms such as the presence of eyelashes or the visualization of the technician's fingers or an image that is too dark with insufficient contrast to assess ischemia. The non-analysis of these peripheral blocks explains the low values of the non-perfusion index found in our study. We were only guided for exclusion of the blocks by the feasibility or not of the non-perfusion evaluation, and in order to reduce bias there were two separate readers that assess the analyzability of the blocks, and to assess the area of non-perfusion on FA.
5) Out of 48 patients, only 25 completed all follow up, and only 20 had FA images that can be used for this study: this high percentage of loss of follow up can cause selection bias. What are the reasons for lost follow up? It is not safe to assume this loss of follow up is random, thus causing bias in the data included. Specifically, in a scenario where the patients who loss follow up because they became more visually impaired to attend appointments, such loss of follow up will neglect a lot of data on people who have worse clinical outcomes, causing bias that is significant enough to compromise the validity of the study. Such concern is not ungrounded: the patient cohort included in this study seems to be healthier in terms of the vision compared to other studies, evident by 1) better baseline vision of 70 ETDRS letters (compared to others at 50s) and smaller percentage of non-perfusion areas. This makes me worry about the generalizability of the study results. Also the number of patients is small.
Thank you for the relevance of your comment. Indeed, only 25 patients had a follow-up to 12 months, the reasons for this loss of follow-up were very diverse, and were analyzed: patient’s decision to stop the study, patient not having respected the protocol schedule, patient having had a change in treatment, patient went abroad during the visit to M12, patients did not return at 12 months for personal reasons, 6 of them were included from 03/2019 to 05/2019 and could not complete their follow-up due to the COVID crisis which started in France in 03/2020, 3 patients had to interrupt follow-up because they were hospitalized for complications other than their diabetic retinopathy (1 for end-stage renal failure and 2 for foot wound). As we can see, the reasons were very diverse and were not only because the patients went less well. Moreover, since it is a monocentric study located in an area of social insecurity, one might assume that patients were less observant (reference 30), which could explain their absences at the M12 visit, in addition to their intense follow-up with other specialists.
Furthermore, statistical analyses did not find any difference in visual acuity or the severity of diabetic retinopathy between patients lost to follow-up (supplementary table added) and those who completed the study.
And despite the absence of difference regarding the non-perfusion index between baseline and M12, we have found a significant improvement in the severity score of diabetic retinopathy which indicates despite the bias that the cohort even small could be analysable.
6) Some minor grammar mistakes: eg 253-254 UWF-FA allows analyzing [not only] the peripheral ischemic retina but also central ischemic area on a single image.
Thank you for your thoroughness, we have corrected this error.
7) Could the baseline characteristics of the 20 interpretable patients be added to the study? Is there any similar analysis done for these 20 patients’ data only (baseline vs M12)? How does that result look like? Given that data of more than half of the original 48 patients were not used in analysis, it remains in question how meaningful of an interpretation can be generated based on the baseline characteristic of all 48 patients.
Thank you for your comment. The description of the 20 patients and the 28 patients not included in analysis has been added in the appendix to the article in a supplementary table. We can observe that the 20 patients have the same baseline characteristics that the 28 patients not included in analysis.
The results presented in Tables 2 and 3 are based only on data from these 20 patients.
|
Variable |
Analysis population |
Exclusion population |
|
N=20 |
N=28 |
|
|
Mean age, years, (± SD) |
63 ± 11 |
68 ± 9 |
|
Male gender, n (%) |
13 (65%) |
18 (64%) |
|
Right eye, n (%) |
13 (65%) |
14 (50%) |
|
Median BCVA, ETDRS letters [IQR] |
75 [59; 77] |
68 [61;75] |
|
Median CMT on OCT, µm, [IQR] |
324 [287; 471] |
357 [310;445] |
|
Type 1 diabetes, n (%) |
0 (0%) |
2 (7%) |
|
Median Hba1c, % [IQR] |
8 [7; 8] |
8 [7;9] |
|
Patients treated with insulin, n (%) |
11 (55%) |
18 (64%) |
|
Macroangiopathy, n (%) |
4 (20%) |
5 (18%) |
|
DR stage, n (%) |
|
|
|
Moderate |
7 (35%) |
12 (43%) |
|
Severe |
13 (65%) |
16 (57%) |
|
Previous treatment for DME, n (%) |
|
|
|
none |
20 (100%) |
25 (89%) |
|
anti-VEGF |
0 (0%) |
2 (7%) |
|
corticosteroids |
0 (0%) |
0 (0%) |
|
laser |
0 (0%) |
1 (4%) |
|
Previous PRP, n (%) |
0 (0%) |
0 (0%) |
|
Lens status: phakic, n (%) |
12 (60%) |
16 (46%) |
|
HBP, n (%) |
7 (37%) |
14 (50%) |
|
Renal status, n (%) |
|
|
|
No renal failure |
16 (84%) |
22 (79%) |
|
Microalbuminuria |
2 (11%) |
1 (4%) |
|
Renal failure |
1 (5%) |
5 (18%) |
|
Dyslipidemia, n (%) |
6 (33%) |
14 (50%) |
|
Sleep apnea, n (%) |
1 (6%) |
1 (4%) |
|
OHT or glaucoma, n (%) |
3 (15%) |
2 (7% |
Reviewer 2 Report
The authors provide an interesting study on the effect of aflibercept on the non-perfusion area using UWF- FA
The manuscript is well written.
There are a few comments:
1. If possible, can the authors provide an image showing how the CNP areas changed from baseline to M12
2. First few lines of the discussion appear to be comments from the previous reviewer. Kindly delete.
3. Was there any discrepancy among graders in characterizing/identifying the CNP areas? (As I can see some doubtful areas in Figure 2 that have not been marked as CNP) If yes, how was this addressed? Kindly mention in methods.
Author Response
The authors provide an interesting study on the effect of aflibercept on the non-perfusion area using UWF- FA
The manuscript is well written.
There are a few comments:
- If possible, can the authors provide an image showing how the CNP areas changed from baseline to M12
This image can be added in the “results” paragraph after line 162.
We have added a figure 4 with an exemple
- First few lines of the discussion appear to be comments from the previous reviewer. Kindly delete.
The lines have been deleted.
- Was there any discrepancy among graders in characterizing/identifying the CNP areas? (As I can see some doubtful areas in Figure 2 that have not been marked as CNP) If yes, how was this addressed? Kindly mention in methods.
Yes, this is specified in lines 110-111, we have completed the method section.
Thank you very much for considering our manuscript.
Best regards,
The authors